# Parallel analysis of tri-molecular biosynthesis with cell identity and function in single cells

Samuel C. Kimmey [1,2], Luciene Borges [2], Reema Baskar [2,3] & Sean C. Bendall [2]

Cellular products derived from the activity of DNA, RNA, and protein synthesis collectively control cell identity and function. Yet there is little information on how these three biosynthesis activities are coordinated during transient and sparse cellular processes, such as activation and differentiation. Here, we describe Simultaneous Overview of tri-Molecule Biosynthesis (SOM$_3$B), a molecular labeling and simultaneous detection strategy to quantify DNA, RNA, and protein synthesis in individual cells. Comprehensive interrogation of biosynthesis activities during transient cell states, such as progression through cell cycle or cellular differentiation, is achieved by partnering SOM$_3$B with parallel quantification of select biomolecules with conjugated antibody reagents. Here, we investigate differential de novo DNA, RNA, and protein synthesis dynamics in transformed human cell lines, primary activated human immune cells, and across the healthy human hematopoietic continuum, all at a single-cell resolution.

[1] Department of Developmental Biology, Stanford University School of Medicine, Stanford, CA 94305, USA. [2] Department of Pathology, Stanford University School of Medicine, Stanford, CA 94305, USA. [3] Cancer Biology PhD Program, Stanford University School of Medicine, Stanford, CA 94305, USA. Correspondence and requests for materials should be addressed to S.C.B. (email: bendall@stanford.edu)

The integrated output of DNA replication, RNA transcription, and protein synthesis imparts gene expression and function in an individual cell. Importantly, the activity of these processes is tightly regulated to maintain tissue homeostasis, or modulated to facilitate changes in cell-state, such as progressing through the cell cycle[1] or differentiation[2]. Much of our collective knowledge of DNA[3], RNA[4–6], and protein[7,8] synthesis in complex systems is derived from labeling cells or tissue of interest with metabolic precursor molecules for a set period of time, followed by fixation and analysis. Conventional approaches to measure these processes use radio-labeled precursor molecules[6,7,9], with newer methods moving toward click-[5,10] or immuno-chemistry[4,6,7] based detection to measure specific synthesis activity in individual cells.

Recent investigations to better understand the regulation of biosynthesis processes in heterogeneous cell populations highlight the strength of layering single-cell activity measurements with parallel quantification of target biomolecules on high-throughput machines[7,8,10–12]. These investigations unified biomarkers informing single cell phenotype and function with their protein synthesis activity[7,10,12], or even proliferative history[8,11], as measured by conventional flow cytometry. In one recent example, investigators characterized the activity of protein synthesis in developmentally isolated hematopoietic populations from mouse bone marrow, establishing a regulated control of protein synthesis activity during hematopoietic cell specification[10,12]. While these studies demonstrate the benefit of measuring protein synthesis activity on single-cell platforms, methodology and reagents to provide parallel assessment of de novo RNA synthesis have yet to be presented. However, their development would provide a unique and novel single-cell dataset unifying cellular biosynthesis activity with cell phenotype and function. Finally, while these recent studies focused on protein synthesis activities in complex cell systems of cell lines and primary mouse tissue[7,8,10–12], there are few reports on comparable human tissue[9,13], those of which utilized radioactive precursors and only reported activity in broad bone marrow morphological groups.

One technical reason inhibiting such studies is the lack of integrated methods that enable fast labeling and robust quantification of de novo molecules of DNA, RNA, and protein, in parallel with simultaneous recording of select biomolecules. The integration of such measurements would allow investigators to probe multiple biosynthesis processes in diverse cell populations with many discrete cell-types or -states by generating multifaceted single-cell datasets, which can be rigorously analyzed in silico. The development of mass-cytometry enabled simultaneous detection of up to 45 distinct biomolecules at a rate up to 1000 cells per second with individually labeled antibody reagents, and importantly does not suffer from technical artifacts of auto-fluorescence or spectral overlap currently present in fluorescent flow cytometry[14–16]. However, one important technical limitation to consider when analyzing cells with mass-cytometry is the inability to sort cells on measured characteristics, as the measurement process is destructive. However, even with its destructive nature, mass-cytometry enables routine measurements of diverse repertoires of biomolecules, yielding thousands to millions of multiplexed single-cell data from a single experiment. The combination of accessible parameter space and sample throughput enable the necessary complexity and depth to capture low-abundant cell types present at frequencies as low as 1 in 10,000[16]. Additionally, the ability to integrate sample-barcoding seamlessly into cell staining steps enables simultaneous staining and analysis of as many as 20 experimental conditions[17], providing robust quantitative comparison and eliminating technical staining variability between individual samples. Thus, we believed this platform would enable robust and parallel assessment of biosynthesis activities and cell biology across diverse cell populations and experimental conditions.

Drawing on recently developed methods to quantify disparate biosynthesis activities and leveraging multiplexed single cell measurement technologies, we developed a simple non-genetic, tri-molecular pulse-labeling strategy to simultaneously quantify the DNA, RNA, and protein synthesis activity of individual cells in a high-throughput manner. A method we termed, Simultaneous Overview of tri-Molecule Biosynthesis, or SOM₃B. Here, we use SOM₃B to provide a detailed overview of DNA, RNA, and protein synthesis in asynchronously dividing cell lines, primary samples of healthy human whole blood, and bone marrow. For each context, we highlight the activity of these processes in individual cells across the cell-cycle, during small-molecule induced cell activation, and across developmentally organized cell phenotypes, respectively.

## Results

**Labeling and detection of nascent DNA, RNA, and protein.** To label nascent biomolecules of DNA, RNA, and protein, we pulsed live cells with an optimized mixture of 5-Iodo-2′-deoxyuridine (IdU), 5-Bromouridine (BRU), and puromycin. Importantly, these three molecules were selected due to their previously established utility in cell activity assays in order to quantify cellular biosynthesis across diverse biological systems (IdU[3,18,19]; BRU[4,6,20–25]; puromycin[7,8,10,11,26,27]). The numerous demonstrated applications of these metabolic pre-cursor molecules on their own, in partnership with multiplex-capable detection systems, such as flow cytometry or immunofluorescence imaging, encouraged us to unify all three in combination in order to provide a snapshot of biosynthesis activities in cell-systems of choice. Further, the potential to obtain detailed information by unifying such metabolic incorporation information with simultaneous biomarkers abundance using multiplexed detection technologies[3,6,7,11,18,19,28], led us to quantify the activity of de novo DNA, RNA, and protein synthesis on a platform, which enablable a high level of combinatorial measurements (i.e. mass-cytometry).

In SOM₃B, each metabolic analogue can enter live cells and is incorporated into actively synthesized macro molecules of DNA, RNA, and protein, respectively (Fig. 1a). After as little as 30 min, the labeling mixture is removed and cells are fixed and stained immediately with metal-isotope labeled antibodies, or fixed and cryopreserved for later sample preparation and analysis. Incorporation of each molecule was quantified on a cell-by-cell basis by mass-cytometry, detecting either elemental iodine for IdU[3] or lanthanide tagged monoclonal antibodies for BRU[6] and puromycin[7] (Fig. 1b–e; Supplementary Fig. 1, see methods).

Along with the concentration of IdU needed to quantify DNA synthesis by mass-cytometry[3], we identified optimal conditions for, and concentrations of, BRU and puromycin to label live cells. HeLa cells treated with increasing concentrations of BRU (Fig. 1b), or puromycin (Fig. 1c), were stained with lanthanide labeled monoclonal antibodies against BRU and puromycin, respectively (detection optimization, Supplementary Fig. 1a). Importantly, selection of optimal concentration of metabolic precursor, and antibody reagents to detect the respective incorporated molecule, is made possible by performing titrations of both classes of reagent on unlabeled and labeled cells. Concentrations of BRU and puromycin were chosen based on observed signal-to-noise ratio in order to provide a large dynamic range of detectable synthesis activities. Concentrations of metal-isotope conjugated antibody used for subsequent SOM₃B experiments were selected by comparing the observed background signal in un-labeled cells with the incorporated molecules in labeled cells (Supplementary

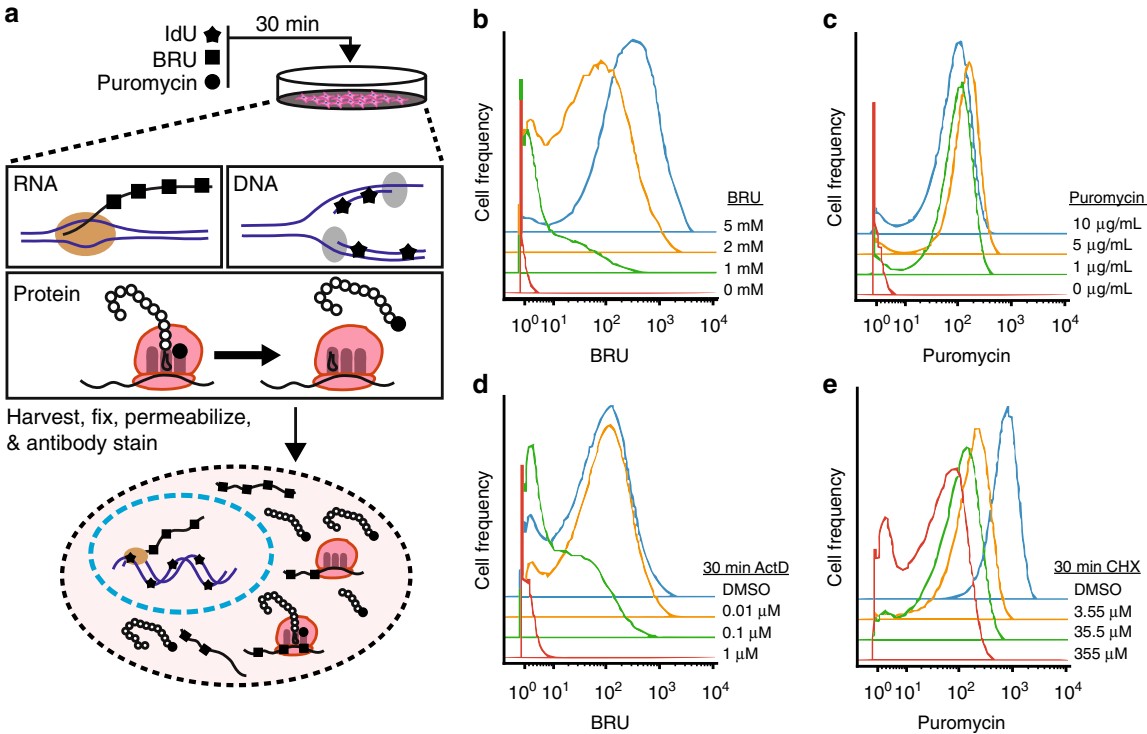

**Fig. 1** SOM$_3$B, simultaneously captures de novo DNA, RNA, and protein synthesis in transient, asynchronous cell systems. **a** Overview of biomolecule labeling with 5-Iodo-2-deoxyuridine (IdU), 5-Bromourdine (BRU), and puromycin incorporation into each biomolecule based on their active synthesis. **b, c** Mass-cytometry quantification of incorporated BRU (**b**) and puromycin (**c**) in HeLa cells pulsed with increasing concentration of each label molecule, detected with isotope labeled antibodies. **d, e** Histogram of BRU and puromycin incorporation in HeLa cells pre-treated with increasing concentration of Actinomycin D (ActD) (**d**) or Cycloheximide (CHX) (**e**), respectively. Experiments were performed multiple times, and representative plots shown

Fig. 1a and Supplementary Table 1). Concentrations of BRU and puromycin were selected based on a combination of previously reported uses of these molecules for labeling purposes[7,11,25], and observation of incorporation compared with unlabeled controls stained with matching concentration of labeled antibody (Fig. 1b, c). Importantly, concentrations of pre-cursor molecule and antibody reagents for new cell types should be optimized and selected to minimize background antibody staining in un-labeled conditions (Fig. 1b, c, red histogram trace).

Notably, inclusion of one or more RNase inhibitors during antibody staining failed to increase overall BRU signal, indicating labeled RNA remains stable and detectable with conventional mass-cytometry staining methods. Interestingly, when compared to sample staining conditions including RNase inhibitors, there is improved detection of incorporated BRU when staining is performed without (Supplementary Fig. 1b). This is likely explained by increased exposure of BRU antigen due to the activity of RNase enzymes. Enzyme-improved detection of incorporated nucleotides is routinely performed when nascent DNA is labeled with BRdU, in which fixed cells are pre-treated with low concentrations of DNase I to expose antigen occluded by compacted DNA or bound protein prior to antibody-detection of BRdU[29,30]. Finally, simultaneous labeling and detection of cells with IdU, BRU, and puromycin did not significantly interfere with quantification or cell signaling activity when compared with single- or double-molecule labeling controls (Supplementary Fig. 1d–f). Altogether, we established SOM$_3$B as a fast and robust strategy to simultaneously capture complementary synthesis activities at the single cell level.

We further assessed specificity and quantification of SOM$_3$B with HeLa cells pre-treated with increasing concentrations of Actinomycin D (ActD) or Cycloheximide (CHX), which block

synthesis of RNA[31] or protein, respectively[32] (Fig. 1d, e). As expected, HeLa cells pre-treated with ActD for only 30 min prior to labeling displayed dose-dependent inhibition of the median BRU incorporation, while the same treatment with CHX also led to a decrease in median puromycin incorporation. While ActD treatment resulted in little change of incorporated IdU or puromycin (Supplementary Fig. 2a), HeLa cells pre-treated with CHX displayed a dose-dependent decrease in median BRU and IdU incorporation, yet maintain a similar proportion of cells actively synthesizing DNA (Supplementary Fig. 2b). These results highlight the dependence of DNA[33] and RNA[34] activity on continued protein synthesis for this cell line, a process previously described with the assistance of radioactive pre-cursor molecules in bulk assays and confirmed here using our SOM$_3$B single-cell approach. Collectively, these results reinforce SOM$_3$B as a simplified approach to capture specific biosynthetic activities by observing predictable changes over large measurement dynamic ranges in individual cells.

**Single-cell biosynthesis dynamics across the cell-cycle.** Insights into cell cycle regulation are compounded by asynchronous biosynthesis activities and cell states, involving numerous regulators. For instance, dividing eukaryotic cells suppress nascent transcription (i.e. RNA synthesis)[1,35] and protein synthesis (up to 35%)[36] during mitosis, relieving suppression after re-entry into interphase[36]. However, our understanding of the activity of RNA and protein synthesis during cell-cycle progression is derived from assays typically involving chemical synchronization and large input material[35,36]. Importantly, this obfuscates the regulatory timing of biosynthesis activity as cells progress through the least frequent (1–2%, Fig. 2a), and shortest phase of the cell

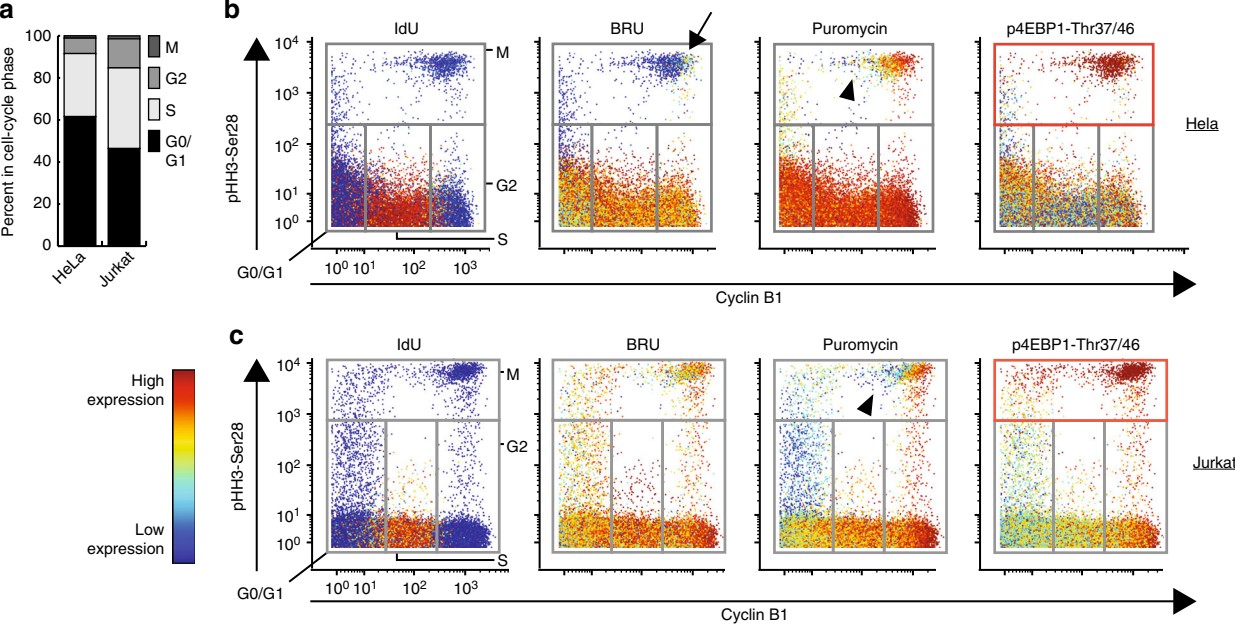

**Fig. 2** de novo RNA and protein synthesis is suppressed in mitotic cells of asynchronously dividing human cell lines. **a** Cell-cycle phase breakdown of HeLa and Jurkat cell lines labeled with SOM$_3$B. **b** Biaxial plot depicting cell-cycle progression of asynchronously dividing HeLa cells overlaid with SOM$_3$B expression and phosphorylation status of 4EBP1-Thr37/46, arrow indicates early mitotic HeLa cells with low de novo RNA synthesis. (CyclinB1neg—G0/G1, CyclinB1mid—S, CyclinB1high—G2, p-HH3-Ser28high—mitotic; expression color range—IdU: 0, 3460; BRU: 0, 697; Puromycin: 1.5, 600; p4EBP1: 0, 73). **c** Biaxial plot depicting cell-cycle progression of asynchronously dividing Jurkat cells overlaid with SOM$_3$B expression and phosphorylation status of 4EBP1-Thr37/46. Arrow head in **b** and **c** mark downregulation of de novo protein synthesis in mitotic cells (CyclinB1neg—G0/G1, CyclinB1mid—S, CyclinB1high—G2, p-HH3 (S28)—mitotic; expression color range—IdU: 0, 676; BRU: 2, 1030; Puromycin: 8.5, 1060; p4EBP1: 5, 114). Experiments were performed multiple times, representative plots shown, 115,000+ cell events displayed for each cell line

cycle, as mitosis lasts only 1 h in HeLa cells[37]. Thus, with this system of a relatively small fraction of total cells occupying a transient cell-state (i.e. mitosis) with previously appreciated discrete biosynthesis activities, we sought to test the sensitivity of SOM$_3$B and quantify dynamic biosynthesis characteristics across a single mixed populations of cell-states. Accordingly, we combined SOM$_3$B with several specific antibodies against cell-cycle specific markers in order to capture and characterize synthesis activity of asynchronously dividing HeLa and Jurkat cell lines[3] (Supplementary Table 1).

As expected and in agreement with recent image-based analysis[35], we observed a marked decrease in HeLa cell RNA synthesis activity during the transition from the G2-phase into mitosis. Mitotic initiation is indicated by increased abundance of phosphorylated histone H3-Ser28[38] (p-HH3 Fig. 2b, arrow). Interestingly, while Jurkat cells also modestly suppressed RNA synthesis activity in mitotic cells relative to G2-phase cells, the degree of suppression is less than that observed for HeLa cells (Fig. 2c). In contrast to RNA synthesis, protein synthesis remained active in early mitotic cells and was down-regulated only in cells at later stages of mitosis (reduced CyclinB1 expression; Fig. 2b, c, arrow head). One mechanism maintaining translation in early mitotic cells is high activity of the mTOR pathway[39], which results in inhibitory hyper-phosphorylation of endogenous translation initiation repressor, Eukaryotic translation initiation factor 4E-binding protein 1, abbreviated as 4EBP1. Indeed, we observed the highest phosphorylation of 4EBP1 (p-4EBP1) in the same early mitotic cells (Fig. 2b, c, red boxes). Altogether, these results establish the capability of combining SOM$_3$B with intracellular markers to capture and visualize regulation of synthesis activity in rare, and transient cells states found within mixed cell populations.

**RNA and protein synthesis in primary peripheral immune cells.** Immune cell activation is a coordinated series of transcription and translation events that can lead to cell division in a context-dependent basis as a result of prolonged stimulation with specific immune-modulatory molecules. Human peripheral blood mononuclear cells (PBMCs) lay in a dormant state until a cell-type specific immune stimulation modulates gene[40] and protein[8] expression, releasing specific soluble signaling factors to further instruct the immune system. Recent investigations in mouse highlight dynamic global protein synthesis rates in primary lymphocyte subsets after 16–48 h of ex vivo cellular activation with immune-stimulatory molecules[11], highlighting both a measureable increase in active protein synthesis and posttranslational modifications of signaling molecules via phosphorylation. Yet, how protein and RNA synthesis is immediately modulated in individual immune cells over the course of short-term cell activation before cell division, or the dynamics of these synthesis activities during human immune-cell activation, remains unexplored.

Thus, we employed SOM$_3$B to directly compare the sequential layers of ex vivo human PBMC activation when stimulated with broad immune-activation molecules, phorbol 12-myristate 13-acetate (PMA) and ionomycin[41], in a time-course up to 5 h (Fig. 3a). Importantly, we employed reagents, which enable instant fixation of nucleated cells, and lyses of red blood cells, in order to capture the time-dependent biosynthesis activity and phosphorylation states as a result of stimulation (see methods). In addition to the discrete layers of RNA and protein synthesis, 15+ phenotypic biomolecules were simultaneously measured in order to assess the unique responses of major immune sub-populations present in whole blood. Importantly, palladium-isotope sample barcoding reagents[17] were applied before pooling fixed cells from

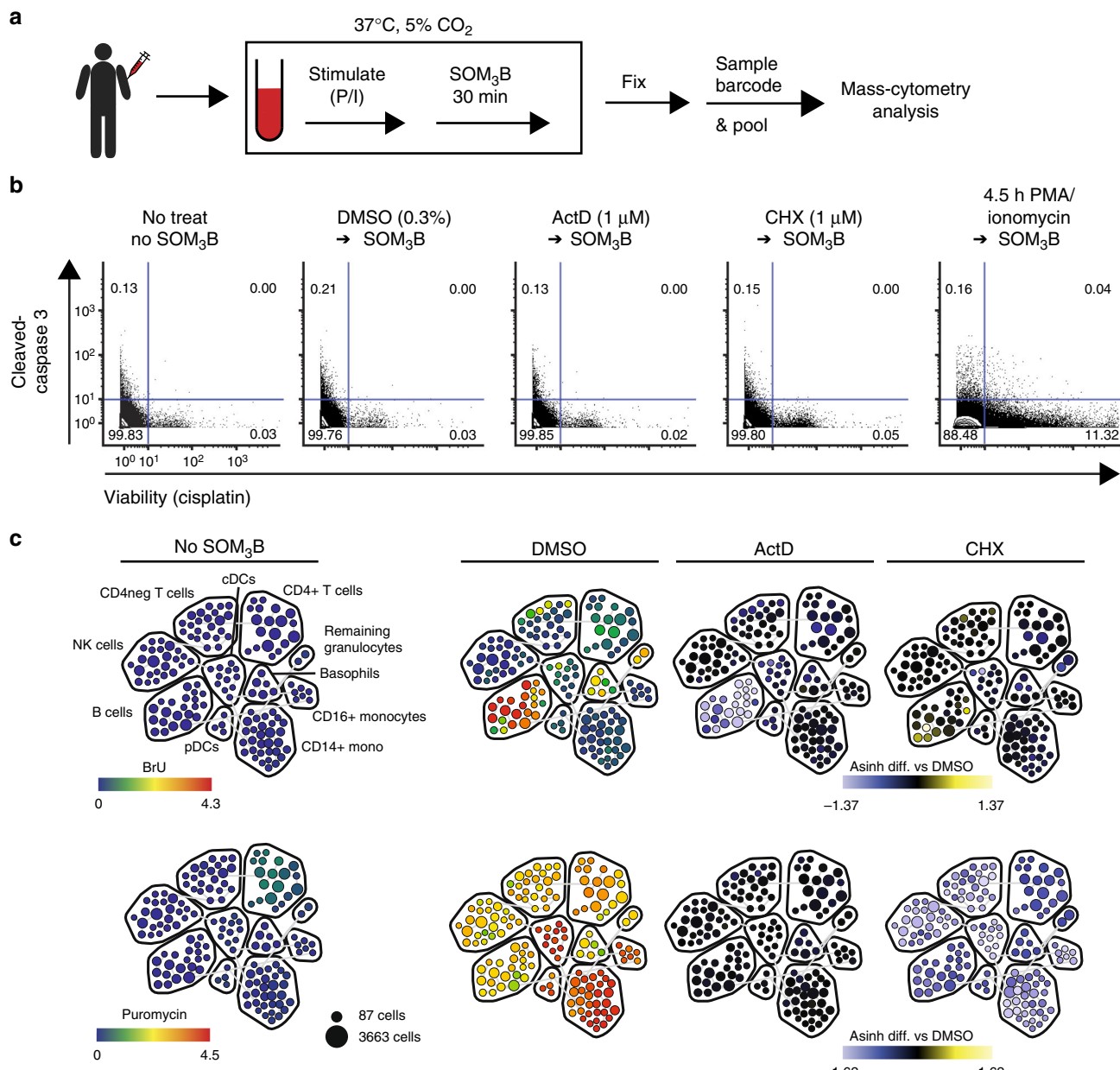

**Fig. 3** SOM₃B does not interfere with primary immune sample cell fitness and predictively reflects biosynthesis activity. **a** Whole blood from a healthy donor stimulated in a time course prior to SOM₃B analysis (P/I; PMA/Ionomycin). **b** Biaxial plots with inlaid frequency of whole blood mononuclear cells undergoing apoptosis (cleaved-Caspase3 positive) and non-viable cells (positive cisplatin staining). Inlaid quadrant numbers indicate the percent of total events within each gate. Approximately 100,000 events displayed for each condition. **c** Manually annotated SPADE map of in silico gated peripheral immune mononuclear cell subsets in whole blood from a single healthy donor not labeled with SOM₃B reagents (far left), or treated with DMSO (middle left), Actinomycin D (middle right), and Cycloheximide (far right) prior to SOM₃B labeling. Rainbow color overlay scaled to indicate median expression for each cluster (BRU—top, Puromycin—bottom). Blue-yellow color overlay scaled to indicate the difference between the transformed median value (ArcSinh cofactor of 5) of a given cluster from the complementary control cluster (DMSO). Weak puromycin signal in "No SOM3B" is derived from background isotopic contamination in CD4+ T cells with high CD4 expression (CD4-Gd157 into Puromycin-Gd158). Number of cell events displayed; 130,000–160,000 single cell events for each map, experiments were performed multiple times, representative plots and maps shown

all time-points into a single tube, followed immediately by antibody staining (Supplementary Table 2).

Focusing first on biosynthetic activities in unperturbed PBMCs, we ensured SOM₃B labeling was robust and did not interfere with cellular fitness or viability (Fig. 3b, c). Importantly, control samples from the same PBMC donor left unlabeled, or pre-treated with specific inhibitors, indicated minimal effect on cell fitness as there were little change in the proportion of cells with activated pro-apoptotic caspase molecules or in cell viability,

examined by antibody staining and cisplatin exclusion[42], respectively (Fig. 3b). For remaining analysis of PBMC experiments, single cell data was restricted to mono-nuclear cells in silico after non-viable and non-mononuclear single-cell events were removed, in addition to excluding granulocytes, red blood cells, and platelets (Supplementary Fig. 3). Mono-nuclear single-cell data from all samples clustered into phenotype-sub groups with SPADE[43] (SPADE column, Supplementary Table 2) illustrates the specificity of SOM₃B when labeling whole blood ex vivo

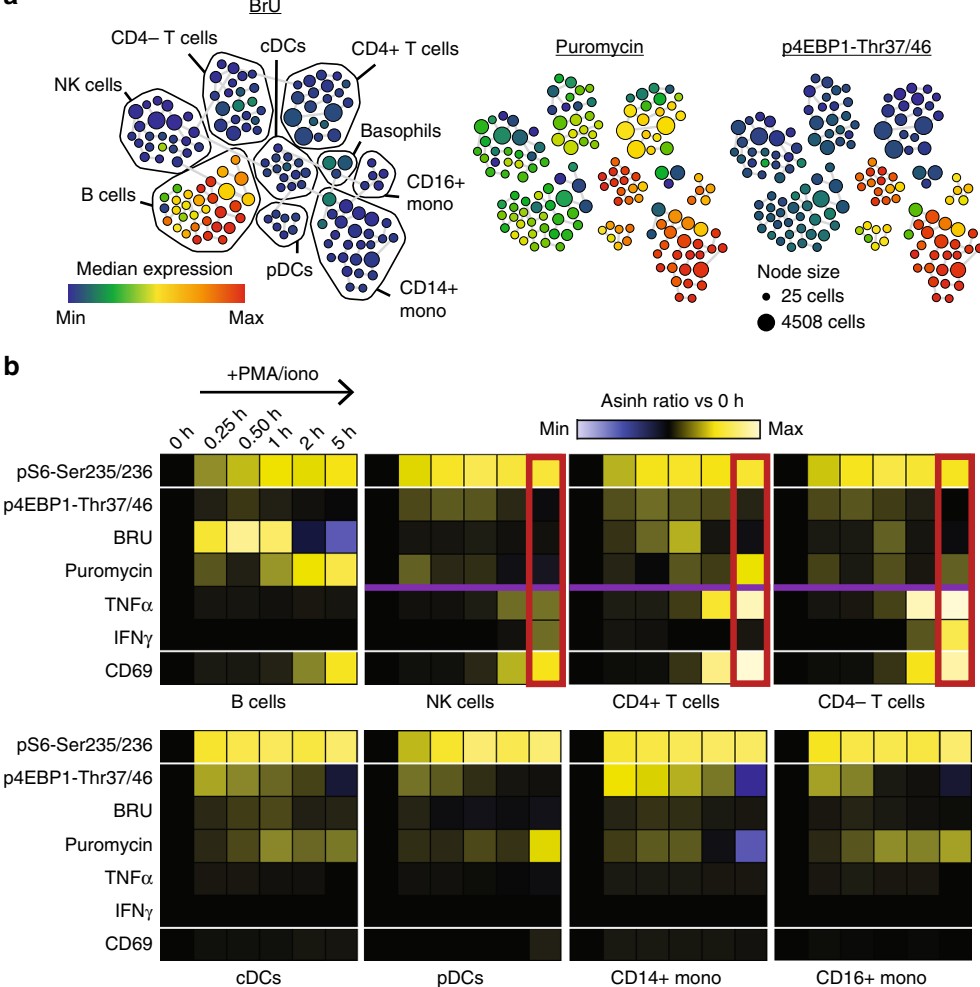

**Fig. 4** Activation of de novo RNA, protein, and cytokine synthesis occurs sequentially during ex vivo peripheral human immune activation. **a** Manually annotated SPADE map of unstimulated, in silico gated, PBMCs clusters, color overlaid with the median baseline incorporation of BRU (left), puromycin (center), and median baseline phosphorylation of 4EBP1-Thr37/46 (right). **b** A heat map summary, ordered by stimulation time and grouped by cell phenotype, of peripheral immune cell subsets with PMA/ionomycin activation, each parameter normalized to 0 h (ratio of transformed cluster median at given time point relative to $t = 0$ h). (ArcSinh ratio color scales; p4EBP1, BRU, Puromycin, TNFalpha, IFNgamma: −1 to 1; pS6, CD69: −3 to 3). Number of cell events displayed; 95,000 to 150,000 mononuclear cell events for each time point, experiments were performed multiple times, with representative maps and plots shown

by predictably blocking SOM$_3$B label incorporation with specific inhibitors (Fig. 3c; cluster phenotype description, Supplementary Table 3).

In SOM$_3$B labeled PBMCs observed at baseline, we find B-cells as the predominant immune cell subset with the highest median RNA synthesis activity while maintaining protein synthesis comparable to other lymphocyte sub-populations. However, myeloid cells including monocytes and dendritic cells maintained a higher median basal activity of protein synthesis facilitated by high activity of the mTOR pathway, indicated by phosphorylation of 4EBP1 (Fig. 4a). Parallel quantification of signal pathway members' post-translational modification (i.e. phosphorylation), downstream modulation of biomolecule synthesis, and cytokine production, enabled robust analysis of the immediate cellular response to PMA/ionomycin (Fig. 4b; Supplementary Table 2).

As expected, PMA/ionomycin stimulated PBMCs, across all identified subsets, immediately (i.e. 15 min) responded beginning with increased ribosomal protein S6 and 4EBP1 phosphorylation, while surface expression of the classical protein activation marker CD69 emerged only with longer stimulation (Fig. 4b, 2 h). Relative to time 0 h, upregulation of the median de novo RNA

synthesis activity occurred during the first hour of stimulation before returning to baseline, followed by activation of de novo protein synthesis and differential detection of cytokines (Fig. 4b, 2 h & 5 h). Interestingly, despite similar cytokine production profiles (Fig. 4b, red box), T- and NK- lymphoid cell subsets had very different protein synthesis kinetics where NK cells more closely resembled the DC and monocyte myeloid cell lineages, indicating global features of their immune response kinetics more aligned with that of the innate immune system (Fig. 4b, purple line). Collectively, these results demonstrated the ability to use SOM$_3$B to uncover timing of biomolecule synthesis activity in coordination with other differential facets of the immune response.

**Dynamic biosynthesis across a human hematopoietic spectrum.** Recent investigations uncovered regulated control of protein synthesis in adult stem cells of mice[10,28] and drosophila[44]. However, the dynamics of protein synthesis, RNA synthesis, and cell proliferation (DNA synthesis) across human adult stem cells and their progeny has yet to be extensively explored in part due to inadequate methods and limitations in obtaining sufficient

amounts of primary sample material. Leveraging our ability to comprehensively phenotype the major developmental stages in human hematopoiesis[41], we sought to probe the biosynthesis characteristics in healthy human bone marrow with SOM₃B.

After we confirmed labeling primary human bone marrow with SOM₃B reagents did not interfere with cellular fitness (Supplementary Fig. 4), we integrated SOM₃B with approaches our our lab pioneered to deeply phenotype developmental progression in human hematopoiesis[41,45] in order to characterize the activity of macromolecule synthesis during human bone marrow homeostasis in two healthy adult donors (Supplementary Table 4). We first removed non-immune single-cell data in silico, yielding over 325 thousand single cell events for each donor (Supplementary Fig. 5). When biosynthesis activity was visualized across all immune cell data in bone marrow datasets, we observed cells predominantly devoted to protein, or RNA synthesis (Fig. 5a, Supplementary Fig. 6a, arrows), while cells actively synthesizing DNA maintained a high activity of protein synthesis and variable transcriptional activity in both donors (arrow head).

To assess the functional association of biomolecular synthesis with cell identity, we organized single cell data by immune biomarkers with SPADE[41], identifying 33 major phenotypic populations (Supplementary Table 5 and 6). We observed an elevated median protein synthesis activity in progenitor cells relative to Hematopoietic Stem Cells (HSCs) across human marrows ex vivo, in agreement with reported results of in vivo protein synthesis in mouse bone marrow[10,12] (Fig. 5b, Supplementary Fig. 6b). We also saw a dramatic increase in both RNA and protein synthesis during progression into the highly proliferative erythroid lineage (Fig. 5c, Supplementary Fig. 6c, gray box). On the opposite end of the spectrum, plasma cells, the memory B-cells tasked with maintenance of humoral immunity (i.e. antibody production), were the most predominate population with high protein synthesis and rRNA expression, yet little detectable de novo RNA synthesis[9] (Fig. 5b, c, Supplementary Fig. 6b, c, arrow). Importantly, statistically evaluating the similarity of multivariate distributions[46,47] of RNA and protein biosynthesis activity between the HSC population and each identified population established significant modulation of these activities during developmental periods of human hematopoiesis, relative to a baseline activity in HSCs (Fig. 5c, Supplementary Table 6). In brief, the similarity of RNA and protein synthesis is measured using FlowMap-FR[46], a recently developed method based on the Freidman-Rafsky nonparametric test statistic[47]. With this method, the distributions of BRU and puromycin incorporation are together evaluated between the HSC population and all other identified cell-subsets within each bone marrow donor dataset. Importantly, control comparisons between two populations sampled from the same phenotypic population always returned the highest probability of similarity (Supplementary Table 6, HSC vs HSC). These results highlight tightly controlled biomolecular synthesis traits intrinsic to cell function and identity across the human hematopoietic hierarchy uncovered by SOM₃B, and further highlight the utility of partnering SOM₃B with high-dimensional analysis tools to interrogate these activities across complex systems.

Homeostasis of B-cells is maintained in the bone marrow by a series of developmentally discrete B-cell identities with unique cell activities, including periods of proliferation and reorganization of signaling network profiles[45]. Leveraging our prior work in establishing single cell developmental trajectories, we used Wanderlust[45] to organize emergent B cells in the marrow to reveal coordination points of DNA, RNA, and protein synthesis activity in the stages immediately before and after immunoglobulin heavy chain gene rearrangement (Fig. 5d, Supplementary Fig. 6d, arrows & gray box). Previously, we demonstrated that this

coordination point lead to rewiring of the B-cell signaling network, highlighted here by S6 activation following the first detectable expression of B cell immunoglobulin heavy chain (IgM)[45] (Fig. 5d, Supplementary Fig. 6d, right arrow). Strikingly, while developmentally flanked by cells displaying bursts of proliferation and translational activity, it was only following the pro- to pre-B cell coordination point that we see a marked increase in RNA synthesis activity for the first time coinciding with the signaling network re-wiring event (Fig. 5d, Supplementary Fig. 6d, BRU). Collectively, SOM₃B identified key coordination points of macromolecule synthesis during B-cell maturation, highlighting a transcriptional burst associated with the re-wired signaling state for the first time in a fleeting fraction of B cells that can be less than 0.01% of the marrow[45].

## Discussion

Here, we described SOM₃B, a simple method to simultaneously capture DNA, RNA, and protein synthesis activity of individual cells in parallel with highly multiplexed single cell analysis. SOM₃B does not require complex cell manipulation, and as established here, can readily be applied to ex vivo primary human samples for investigation of heterogeneous behavior in complex tissues. We demonstrated several applications of SOM₃B, including the detailed analysis of biosynthesis activities in human cell line cell-cycle continuums, primary human immune cells, and across the hematopoietic hierarchy of healthy human donors. Importantly, the sensitivity and range of measureable activity captured with SOM₃B was established by using a combination of specific inhibitors and the investigation of cell states with previously documented activities of these processes measured with orthogonal approaches in human cell lines. These validation experiments encouraged us to apply SOM₃B to more complex cell populations, including primary human samples.

While single cell activity measurements reported here are the first datasets produced using SOM₃B, the presented data on human immune cell activity can be informed and supported by recent investigations in mouse and historical investigations in human. In mouse, Signer and colleagues[10,12] observed a regulated protein synthesis rate across developmentally linked sub-populations, first highlighting a reduced rate of protein synthesis in hematopoietic stem cells relative to whole bone marrow and additional identified progenitor populations, an observation also present within our human datasets. Single cell analysis of protein synthesis in mouse lymphocytes also illustrated an increase in protein synthesis activity after exposure to stimulatory molecules[7,11] or virus[8]. Finally, the erythroid compartment identified within our human bone marrow datasets displayed a progressive increase in the frequency of cells synthesizing DNA, RNA and protein. Likewise, the mouse erythroid developmental trajectory from early to late progenitors is characterized by a positive association of cell-cycle remodeling, demonstrated by a steady increase in the proportion of cells actively synthesizing DNA in progressive erythroid developmental subsets in fetal liver[48] and bone marrow[49].

Past investigations of human bone marrow cell metabolism obtained with radioactive pre-cursor molecules corresponds with results obtained with SOM₃B and presented here. Incorporation of radioactive precursor molecules illustrated high activity of DNA and RNA synthesis in immature bone marrow cells relative to mature cell types[13], consistent with our results with SOM₃B in which there was no detectable DNA and very little de novo RNA synthesis in mature lymphocyte populations. Further, analysis of radioactive amino acid incorporation in distinct bone marrow cells illustrated a high activity of protein synthesis in immature cell types relative to mature immune cells, similar to the activity

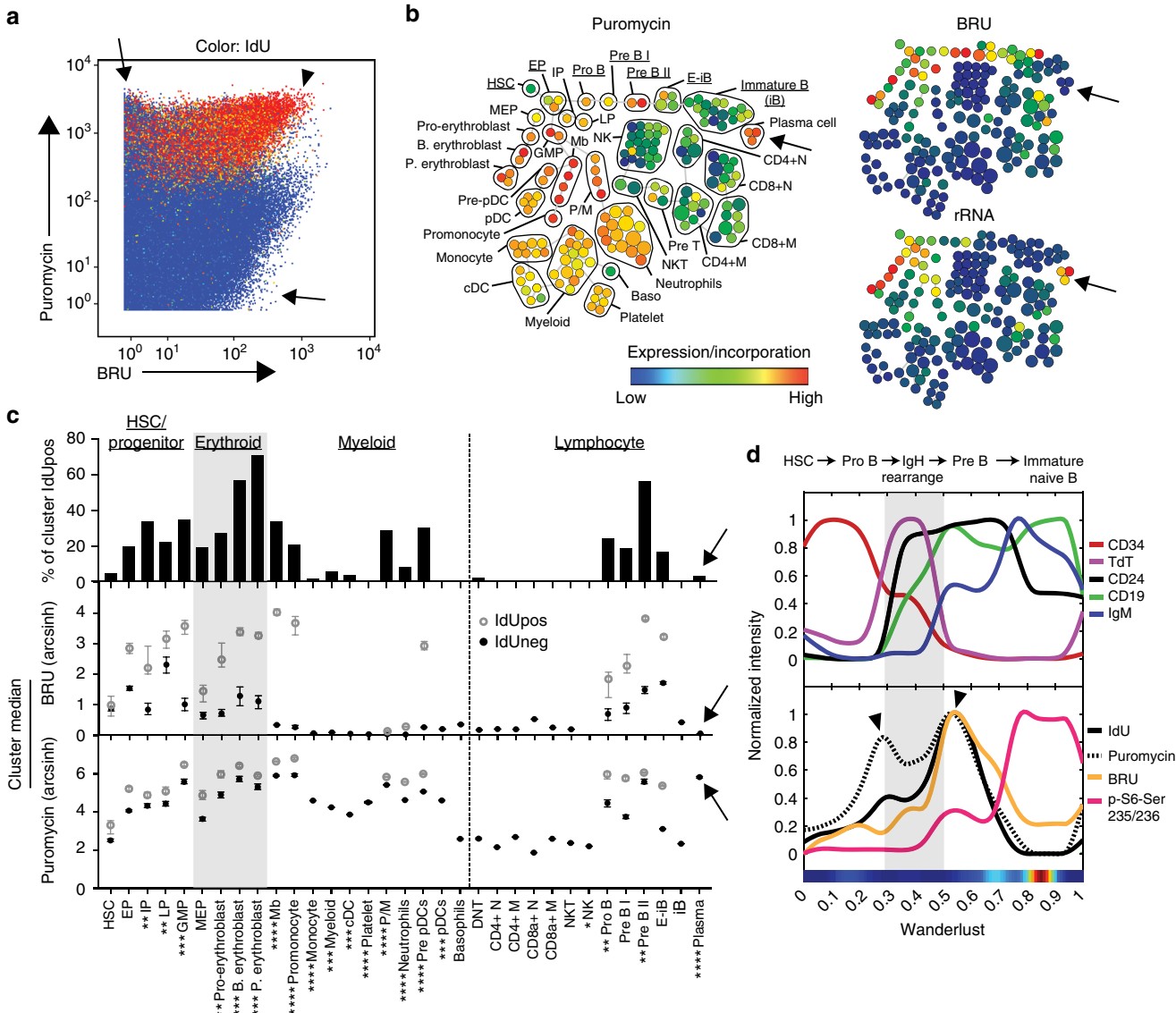

**Fig. 5** Dynamic DNA, RNA, and protein synthesis across a human hematopoietic continuum can be linked to cellular function and developmental identity. **a** Plot of CD45pos immune cells from a healthy human bone marrow donor (color overlaid: IdU incorporation, color range: 0, 410 ion counts). Arrows highlight cells with high activity of RNA or protein synthesis, arrowhead indicates high biosynthesis activity of all three biomolecules. **b** Manually annotated SPADE map of single cell data in **a**, clustered according to immune-cell phenotype and color-overlaid by the median incorporated puromycin (left), BRU (top right) and ribosomal RNA expression (bottom right) for each cluster. **c** Quantification of DNA synthesis (percent IdU positive), RNA and protein synthesis for each phenotype group in **b**. Values are displayed for both IdU positive and IdU negative sub-populations only when greater than 4% of the parent population is positive for IdU. Error bars, 95 percent bootstrap confidence interval, see Supplementary Table 6 for total cell observations in each phenotypic group. Arrows highlight plasma cells. **d** Wanderlust trajectory analysis of B-cell lineage (underlined in **b**) ordered according to developmental pseudo-time with functional markers (top), and activity of DNA, RNA, protein synthesis, and p-S6 expression in the same B-cell developmental axis (bottom). Arrowheads highlight developmental stages immediately before, and after, B-cell somatic recombination. Experiment was performed on two independent donor bone marrow samples, with data from the additional donor presented in Supplementary Fig. 6. p-values symbols expressed as follows: $1.0 \times 10^{-7} > p > 1.0 \times 10^{-14}$*; $1.0 \times 10^{-14} > p > 1.0 \times 10^{-21}$**; $1.0 \times 10^{-21} > p > 1.0 \times 10^{-28}$***; $p < 1.0 \times 10^{-28}$****. Significance was obtained by the Friedman-Rafsky test, evaluating BRU and puromycin incorporation in a given phenotypic population relative to HSC

of DNA and RNA synthesis[9,13]. Importantly, a consistent trend of maintained protein synthesis in the absence of strong RNA synthesis activity in lymphocyte cell types is observed in experiments with radioactive pre-cursor molecules[9] and here with SOM₃B reagents. Taken together, the collective characteristics of biosynthesis activities previously reported in a spectrum of mouse[7,8,10–12,48,49] and human[9,13] immune cells support results presented here using SOM₃B.

Here we demonstrate the utility of SOM₃B to create regulatory models of sophisticated cell systems. An important consideration

when using SOM₃B is the adequate optimization of reagents for the cell type of interest, and inclusion of control conditions including matched, un-labeled cells and/or cells pre-treated with small molecule inhibitors of the biosynthesis processes themselves. This will enable identification of thresholds to set for IdU incorporation and determination of background signal derived from technical artifacts or non-specific binding of antibodies. We envision its combination with prospective isolation schemes for the timing and targeting of cell populations with unique macro-molecule synthesis profiles, followed by orthogonal molecular

profiling and affinity-based purification methods to facilitate de novo discovery of actively regulated molecules (e.g. BRU-seq[25]). Such partnerships will not only provide a profile of molecules contributing to activities measured with SOM3B, but also provide an avenue to address molecular underpinnings driving cell-type specific biosynthesis activities. Importantly, while SOM3B is presented here in partnership with mass-cytometry to enable direct measurement of nascent DNA[3], and robust multiplexing of molecular targets and sample conditions[17,41], use of metabolic pre-cursors and detection reagents optimized for fluorescent detection (e.g. fluorescently labeled antibodies against BRU[4,6] and puromycin[7,11], click-chemistry assisted DNA[50] or RNA[5] synthesis activity, etc.) will allow parallel assessment with less comprehensive, but widely used, detection systems such as fluorescent flow cytometry. Moreover, SOM3B can conceivably be combined with newly developed single-cell sequencing methods (i.e. CITE-seq), which can quantify both endogenous mRNA and nucleotide tagged antibodies[51]. Finally, we expect that SOM3B should be extendible to animal models, enabling in vivo biosynthesis activity approximations in disease models and various genetic backgrounds, altogether highlighting multiple applications of SOM3B in different areas of life science research.

## Methods

**Labeling molecules.** 5-Iodo-2-deoxyuridine (Sigma I7125) was re-suspended in DMSO (Sigma D2650) at 500 mM, 5-Bromouridine (Sigma 850187) was re-suspended in Dulbecco's Phosphate-Buffered Saline (DPBS, Thermo Fisher 14190250) at 100 mM, and puromycin (P212121 58-58-2) was re-suspended in double distilled water at 1 mg/mL. Solutions were gently heated in a 50 °C water bath to completely dissolve solute when necessary. To label nascent DNA, RNA, and polypeptides, all three label molecules were added together and mixed thoroughly with media from actively dividing cells (final concentration in media: IdU; 100 μM, BRU; 2–5 mM, puromycin; 1–10 μg/mL), and then returned to the culture dish for a labeling duration of 10–30 min. Thresholds for cells actively synthesizing DNA were determined by inclusion of unlabeled controls from the same cell type.

**Cell culture.** HeLa (ATCC CCL-2) and Jurkat (ATCC CRL-2899) cells were maintained in DMEM and RPMI (Gibco), respectively, at 37 °C 5% CO2. Human Embryonal Kidney (HEK 293T, ATCC CRL-3216) cells used for puromycin antibody titrations were cultured in the same manner as HeLa cells. Cell culture media for both HeLa and Jurkat cell lines was supplemented with 10% Fetal Bovine Serum (Sigma) and GlutaMAX supplement (Thermo Fisher). For inhibitor experiments, Cycloheximide solution (Sigma C4859) was added directly to cell culture media at the indicated final concentrations. Actinomycin D powder (Sigma A1410) was dissolved in DMSO at 1 mM, and was also added directly to cell culture media at the indicated final concentrations. After 30 min of inhibitor treatment, a mixture of all three labeling molecules was added directly in the same culture media containing inhibitor and returned to the incubator for 30 more minutes before harvesting.

To harvest, HeLa cells were washed with DPBS twice, incubated with Accumax (Thermo Fisher) for 3–5 min at 37 °C, and subsequently mechanically dissociated into single cell suspension with a P1000 micropipette. Jurkat cells were pipetted thoroughly with a P1000 micropipette to dissociate cell clumps.

Single cell suspensions were centrifuged at room temperature for 5 min, 250G, and supernatant aspirated. Cell pellets were re-suspended with cisplatin for 5 min at room temperature (Sigma, 0.5 μM final concentration in DPBS), to label non-viable cells[42]. Cells were next washed with DPBS, and fixed with 1.6% paraformaldehyde (Electron Microscopy Sciences, diluted in DPBS), for 10 min at room temperature. Fixed cells were washed with DPBS, re-suspended in Cell-Staining Medium (CSM: PBS with 0.5% BSA and 0.02% NaN3) and immediately continued to staining with metal-isotope antibodies, or stored long term at −80C. All centrifugations of fixed cells were 5 min, 550 × g at 4 °C.

**Blood donors.** All human blood and bone marrow samples were obtained and experimental procedures were carried out in accordance with the guidelines of the Stanford Institutional Review Board. The experimental procedures / protocols in combination with the samples used in this study were approved under Stanford IRB protocol #42195, which was reviewed by Stanford IRB panel IRB-98.

**Ex vivo stimulation of primary human peripheral blood.** Four heparinized tubes of whole blood from a single healthy donor was obtained from the Stanford Blood Center on the same day it was collected from the donor. Immediately after receiving, whole blood was equally distributed into 10 tubes, and all were placed in a 37 °C 5% CO2 incubator for 30 min before stimulation.

After resting whole blood for 30 min at 37 °C, brefeldin A solution (BioLegend) was added to each stimulation tube to block membrane vesicle transport and permit intracellular detection of induced cytokine protein expression, including the "No stimulation/0 h" control sample. At the same time, PMA (Sigma, 50 nM) and Ionomycin (Sigma, 1 μg/mL) was added to the 5-hour stimulation sample ($t = 0$ h), with the remaining tubes receiving PMA and Ionomycin at set intervals to yield a time-course of cell activation (stimulation duration: 0, 0.25, 0.5, 1, 2, 5 h). A pre-mixed solution of IdU (100 μM), BRU (5 mM), and puromycin (10 μg/mL) was added to all tubes at t = 4.5 h and mixed thoroughly. Cisplatin (1 μM) was added for the final 5 min to stain non-viable cells. At t = 5 h, Phosflow Lyse/Fix Buffer (BD Biosciences, Cat. #558049) was added to all six tubes to lyse red blood cells and simultaneously fix mononuclear cells, including granulocytes and platelets, which are removed from downstream analysis in silico (Fig. S3, Table S3). After fixation at room temperature for 15 min, cells were washed with DPBS twice, re-suspended in CSM and stored at −80 °C until antibody staining.

In addition to the stimulation time-course, remaining tubes of whole blood from the same donor were treated with either (1) 1 μM cycloheximide, (2) 1 μM actinomyocin d, or (3) DMSO for 30 min, followed by addition of a pre-mixed solution of IdU, BRU, and puromycin for another 30 min (same concentration as stimulation tubes). The final tube was left un-treated and did not receive any labeling molecules to serve as an unstimulated, non SOM3B labeled control. As with the stimulation tubes, all four tubes received Phosflow Lyse/Fix Buffer (BD Biosciences, Cat. #558049) at the same time, were then incubated at room temperature for 15 min, and were washed twice with DPBS and stored in CSM. Prior to antibody staining, $5 \times 10^6$ cells from each of the 10 aforementioned sample conditions were metal-isotope barcoded with palladium isotopes[17] and subsequently pooled into a single tube for antibody staining.

**Ex vivo labeling human bone marrow.** Fresh bone marrow aspirates from two independent donors were obtained from AllCells (Alameda, CA) on the same day of collection, and immediately after receiving was placed in 37 °C 5% CO2 incubator for 30 min prior to SOM3B labeling. A mixture of all three label molecules was added together and mixed thoroughly (final concentration; IdU—100 μM, BRU—2 mM, puromycin—10 μg/mL), and then added to pre-warmed bone marrow and returned to 37 °C 5% CO2 incubator for 30 min.

Mono-nuclear cells were subsequently isolated from whole bone marrow with Ficoll to remove granulocytes and erythrocytes, following manufacture instruction (GE Life Sciences, 17144003), and fixed with 1.6% paraformaldehyde diluted in DPBS for 10 min at room temperature, washed with DPBS, and stored in CSM at −80 °C until antibody staining.

For antibody staining, $10 \times 10^6$ cells from both donors were barcoded[17] and pooled in a single tube for antibody staining.

**Antibody staining and data acquisition.** All metal-isotope labeled antibodies used in this study were conjugated and validated as previously described[3,41,52,53], using the MaxPar X8 Antibody Labeling kit per manufacturer instruction (Fluidigm), or were purchased from Fluidigm pre-conjugated. A list of staining concentrations and manufacturer information for each antibody used is found in Supplementary Tables 1, 2 and 4. Purified anti-BRdU (BD, Cat# 555627) was in-house conjugated and used to detect incorporated BRU, as was purified anti-Puromycin (MilliporeSigma, Cat# MABE343). Each conjugated antibody was quality checked and titrated to optimal staining concentration using a combination of primary human cells and/or cancer cell lines. Optimal concentrations of antibodies used in this study were selected based on visual inspection of performed titration on positive and negative cell types, with optimal concentrations demonstrating high signal-to-noise ratio.

Cell staining was performed as previously described for mass-cytometry analysis[3,45,52]. Briefly, fixed cells stored in CSM from a single sample, or barcoded[17] and subsequently pooled cells, were washed with CSM once then centrifuged to pellet cells. For cells stained in the presence of RNase inhibitors, inhibitors were added to every buffer used at their indicated concentration beginning at this point. A mixture of all isotope-labeled antibodies against extracellular antigens was filtered (Ultrafree centrifugal filter—0.1 μm) to remove any precipitate. The filtered antibody cocktail was added to cells suspended in a total volume of 100 μL and incubated at room temperature for 30 min. Surface-stained cells were then washed once with CSM, and permeabilized with ice cold methanol (Sigma), on ice for 10 min. Membrane permeabilized cells were washed twice with CSM, and all isotope-labeled antibodies against intracellular antigens were pre-mixed and filtered before staining in the same fashion as surface antibodies. After intracellular antibody staining, cells were washed once with CSM and then re-suspended in DNA intercalator solution (1.6% PFA diluted in PBS, 1:5000 DNA intercalator from Fluidigm) until ready for data acquisition on a CyTOF 2. This duration was for at least 20 min at room temperature when samples were analyzed by mass-cytometry on the same day, or overnight at 4 °C for acquisition on the day after sample staining.

To prepare cells for acquisition on a CyTOF 2, cells in DNA intercalator were washed once with CSM, and then twice in double distilled water (ddH2O). EQ Four element calibration beads (Fluidigm) were diluted in ddH2O 1:10, and were subsequently used to re-suspend stained cells (~10^6 cells/mL) to be injected into the CyTOF 2. Single cell events were recorded at an event rate of ~500 cells/second.

**Mass-cytometry data analysis**. After data acquisition, FCS files with single cell data were normalized, de-barcoded, and transformed using the inverse hyperbolic sine (ArcSinh) function with a cofactor of 5[52]. Non-viable (cisplatin positive) and actively apoptotic (cleaved-PARP and/or cleaved-Caspase3 positive) cells were removed for all subsequent data analysis. Histograms, biaxial plots, heat plots, and SPADE clustering were created and performed, respectively, using cytobank.org. All data presented for experiments performed on transformed human cell lines contain 90,000–100,000 single cell events per experimental condition unless otherwise noted. All data presented for primary peripheral blood contains 95,000 to 170,000 viable, non-apoptotic single cell events per experimental sample unless otherwise noted. Gating thresholds for IdU incorporation were set based off of unlabeled-control cells. Importantly, these thresholds should be determined by performing a control experiment with un-labeled cells for SOM$_3$B experiments performed on new cell types and/or systems, as some cell lines display variable background incorporation of IdU.

Bootstrap median and confidence intervals, and Friedman-Rafsky (FR) statistics, generated for Jurkat cell data presented in Supplementary Fig. 1e, f were obtained using the same parameters as performed with bone marrow data (further explanation below). For each class of incorporated molecule (i.e. IdU, BRU, and puromycin), an FR probability statistic was generated by comparing single-cell distribution for a given molecule in combination with p4EBP1(Thr37/46) expression, between the indicated experimental condition to the 'IdU + BRU + Puro' sample (IdU incorporation was evaluated using only IdU positive events). This generates a probability value indicating similarity of the incorporated molecule and 4EBP1 phosphorylation expression distributions in Jurkat cells labeled with all three molecules compared with double- or single-label controls. See section below on bone marrow data analysis for further detail and relevant references.

For peripheral blood experiments, mononuclear cells were isolated in silico (Supplementary Fig. 3), and subsequently clustered by cell phenotype with SPADE (down-sampled events target: 15 percent, target number of nodes: 150). Meta-clusters were annotated using established phenotypic markers (Supplementary Table 3), and marker expression (or molecule incorporation) in each population of interest was determined by calculating the difference in the ArcSinh transformed median values for a given time point, relative to the corresponding population in the no-stimulation control sample.

For analyzed bone marrow cells, CD45 positive cell events (Supplementary Fig. 5) were clustered by cell phenotype with SPADE (down-sampled events target: 15 percent, target number of nodes: 175). Meta-clusters were annotated using established phenotypic markers (Supplementary Table 5), and cell events within each phenotypic cluster were exported from cytobank as individual FCS files and subsequently analyzed in RStudio (Version 1.0.153). The frequency of cells with greater than 10 counts of incorporated IdU was used to quantify DNA synthesis activity for each immune-cell phenotype cluster, determined by visual comparison between unlabeled and IdU labeled samples. Bootstrap statistics were generated using boot[54,55] and flowCore[56] libraries. Ordinary Nonparametric Bootstrap was used and replicated 1000 times ($R = 1000$) for each immune-cell phenotype cluster to obtain the bootstrapped median values and 95% confidence intervals for each analyzed phenotypic cluster, with sampling from the raw single cell data of each assigned phenotypic cluster. Supplementary Table 6 displays the total number of cell observations in each phenotypic cluster presented in Fig. 5 and Supplementary Fig. 6. Graph figures were created in Prism (graphpad).

Statistical testing to evaluate the equivalence of BRU and Puromycin incorporation between the Hematopoietic Stem Cell (HSC) population relative to each additional bone marrow populations was evaluated using FlowMap-FR, of the flowMap library[57], implemented in R. FlowMap-FR[46] is an analytical and statistical tool used to compare multivariate single-cell expression data between cell populations. In brief, a pooled sub-dataset is made containing an equal number of single cell events from the two populations under comparison. Next a minimum spanning tree (MST) is generated using the Euclidean distance between all cell pairs in the pooled sub-dataset with respect to BRU and puromycin incorporation. The Friedman-Rafsky[47] statistic is derived from the extent of single-cell data mixing in the resultant MST between both populations, by evaluating the number of edges in the MST connecting a pair of cells that were not derived from the same cell population. Creation of the sub-dataset and MST generation is iterated in order to obtain a probability of population equivalence, with probabilities closer to 1 indicating that populations are likely drawn from the same distribution and small values (i.e. $p < 1.0E$-7) indicating the populations are likely not derived from the same distribution (i.e. population). For our analysis, sampling parameters $S = 200$ (total number of cells in the sub-dataset used generate the MST) and $N = 100$ (total number of iterations) were used in calculating the p-values presented for each donor in Supplementary Table 6[46]. Increasing parameters to consider a larger sub-dataset ($S = 400$), or more iterations ($N = 500$), lead to a general and consistent decrease in $p$-value scores across all populations, except for control HSC vs. HSC, as expected[46].

Wanderlust[45] was used to organize single cell events along the B-cell developmental trajectory using previously characterized phenotypic markers, implemented through the matlab GUI, CYT (Dana Pe'er Lab). Cells belonging to the B-cell developmental trajectory were concatenated into a single FCS file before importing into CYT. Cell parameters used to construct the linear trajectory are found in Supplementary Table 4.

**Reporting summary**. Further information on experimental design is available in the Nature Research Reporting Summary linked to this article.

## Data availability
Single cell mass-cytometry datasets generated and presented for primary human peripheral blood (FR-FCM-ZYR5) and bone marrow (FR-FCM-ZYQV) experiments are available in flowrepository.org.

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

## Acknowledgements

S.C.K. is supported by the NIH/NIGMS Cell and Molecular Biology Training Grant (T32GM007276). R.B. is supported by the A*STAR National Science Scholarship (PhD). S.C.B. is supported by the Damon Runyon Cancer Research Foundation Fellowship (DRG-2017-09), the NIH 1DP2OD022550-01, 1R01AG056287–01, 1R01AG057915-01, 1-R00-GM104148-01, 1U24CA224309-01, 5U19AI116484-02, U19 AI104209, The Bill and Melinda Gates Foundation, and a Translational Research Award from the Stanford Cancer Institute.

## Author contributions

S.C.K. and S.C.B conceived and designed the study, S.C.K. performed all experiments and data analyses. L.B. performed peripheral blood and bone marrow experiments and data analyses. R.B. assisted with statistical analysis. S.C.K. and S.C.B wrote the manuscript.

## Additional information

**Competing interests:** The authors declare no competing interests.

