## [Peer Review File · Nature Communications]

Reviewers' comments:

Reviewer #3 (Remarks to the Author):

The authors have presented the first demonstration of multiplexed measurement of DNA replication, RNA transcription, and protein synthesis alongside tens of additional biomolecule targets via pulsed labelling and antibody probing, all measured in parallel from single cells using mass cytometry. The impressive multiplexing enabled by mass cytometry and the authors' labelling strategy creates a useful tool to understand cellular dynamics at the single-cell level, and the authors have demonstrated its utility by showing its application to understanding cell cycle dynamics of human cell lines as well as stimulated whole blood mononuclear cells and the hematopoietic continuum from clinical samples. While this kind of labelling technique has the potential to open up new studies understanding regulation processes in biological systems, the paper would benefit from additional characterization/validation of the technique in order to improve its adoption and reproducibility by other researchers.

The major claim of the paper is the first demonstration of simultaneous single-cell measurement of de novo DNA, RNA, and protein molecules, alongside multiplexed detection of a suite of other biomolecules, allowing insight into biomolecule synthesis and dynamics. Claims are novel and paper would be of interest. This kind of multiplexed measurement will aid in the design of studies to better understand how de novo DNA, RNA, and protein synthesis is related to phenotype and cell state across biological systems, as well as the dynamics of cell response to stimulus. Several of the points raised by the prior reviewers in the previous version of the manuscript (specifically related to quantitative analysis of the data: prior Reviewer 3's question regarding whether a bootstrapping approach like that used in Fig. 3 to estimate confidence intervals could also be used across the remaining datasets of the work). The paper would benefit from additional quantitative analysis to support some of the claims (please see suggested revisions below). This quantitative analysis would strengthen the key claims of the paper and improve adoption and reproducibility by other researchers. A clearer description of what has and has not been previously characterized and validated with respect to the labelling molecules used in this work would, along with the authors' own characterization, demonstrate solid validation of the technique.

While the manuscript is clearly written, some sections could benefit from additional explanation to aid in understanding by and impact on the broad audience of this journal (e.g.: previous work on and validation of these labeling molecules; capabilities and limitations on quantification and comparisons between different sample and cell types using this labelling technique). Additional statistics and quantification to back up key claims would be beneficial (see suggestions below). A brief overview of the specific labelling molecules used in the work (or very similar molecules) and their previous application would improve both the completeness of previous literature citation as well as background basis for the work (e.g. what validation of these techniques and molecules has already been performed by others vs. what needed to be performed in this contribution). This could be included either in the background section or at the beginning of the results section, where the molecules are introduced.

Suggested revisions:

1. As the primary contribution of this work is the tri-molecular labelling and quantification strategy for single cells, the reproducibility and quantitative detection thresholds for this strategy need to be assessed. I could not see any sample-to-sample measurement variability metrics outlined in the manuscript, and an understanding of this variability is crucial to drawing conclusions from the data

presented. Detection thresholds for significant changes in measured metrics need to be defined in order to conclude that “simultaneous labeling and detection of cells with IdU, BRU, and puromycin did not interfere with quantification or cell signaling activity when compared with single- or double-molecule labeling controls”, that “Bone marrow cell fitness and surface protein expression are unperturbed by labeling with SOM3B reagents”, or that “SOM3B does not interfere with primary immune sample cell fitness”. Indeed, the plots and presented data suggest these conclusions, but the authors’ claims would be strengthened by a quantitative analysis showing that any changes are within the measurement variability of the system. In several figures (e.g. Figs. 1, 2, 3, 4, S1, S2) the authors state that experiments were performed multiple times, but statistical analysis of this replicate data would support the authors’ conclusions and strengthen their validation of this technique.

a. The data of Fig. S1 (crucial validation of the ability to perform the simultaneous measurements key to the claims of this work) would benefit from a quantitative statistical analysis to show that each measurement does not interfere with the others, or with cell signaling measurements (to support the statement “simultaneous labeling and detection of cells with IdU, BRU, and puromycin did not interfere with quantification or cell signaling activity when compared with single- or double-molecule labeling controls”. The inhibitor validation data of Fig. S2 would also benefit from a similar quantitative analysis of dose-dependent effects (while the authors describe changes in the median BRU/puromycin incorporation I did not see any quantification of these medians or their uncertainty). These additional quantitative assessments would provide key validation for the primary novel contribution of this work.

b. The data of Fig. 3 would also benefit from quantitative statistical analysis to support the claim of “control samples from the same PBMC donor left unlabeled, or pre-treated with specific inhibitors, indicated minimal effect on cell fitness as there were little change in the proportion of cells with activated pro-apoptotic caspase molecules or in cell viability, examined by cisplatin exclusion” (p.9). A similar comment for p.10 (Fig. S4): “we confirmed labeling primary human bone marrow with SOM3B reagents did not interfere with cellular fitness (Fig. S4)...”.

2. In Fig. 2 the IdU colour ranges for the two cell lines differ dramatically (~5x greater max for the HeLa than the Jurkat). Does this allow us to draw any information about the cell types in question, or is higher labelling with the three molecules expected for some cells compared to others for reasons unrelated to cellular synthesis rates? Does this present any implications for quantification of diverse clinical sample populations or assay optimization?

3. P.6: The authors state that the data of Fig. S2, showing an effect of cycloheximide treatment on transcription and DNA synthesis, suggest “a dependence of DNA and RNA activity on continued protein synthesis for this cell line”. Additional support (by referencing of past studies investigating effects of cycloheximide on transcription and DNA synthesis) would help support this conclusion on the mechanism driving the observation.

4. For the stimulation experiment, should there be a control for the time between blood draw/brefeldin A treatment (permitting intracellular detection of cytokines) and initiation of stimulation? If not, please provide a rationale and reference(s).

5. Additional experimental details would be beneficial to ensure reproducibility and adoption of this technique:

a. On page 5, the authors state: “After as little as 30 minutes, the labeling mixture is removed and cells can be fixed or cryopreserved for later analysis”, but cryopreservation prior to fixation is not described in the methods (methods state “After fixation at room temperature for 15 minutes, cells were washed with DPBS twice, re-suspended in CSM and stored at -80C until antibody staining.”. Were some samples cryopreserved prior to fixation? If so, in what types of experiments is this the preferred strategy vs. immediate fixation?

b. The methods should include product numbers, particularly for the antibodies in Tables S1 and S2. Although manufacturer, target, and clone of each antibody are all presented in these tables, specific product numbers and potentially lot numbers would be beneficial for reproducibility of this antibody-reliant study.

c. The authors should describe how the optimal antibody concentrations were chosen from the data in Fig. S1a-b. Similarly, a brief summary of how the optimal BRU/puromycin concentrations were chosen based on the data of Fig. 1 b-c (and potentially whether there is any variation in

optimal concentration based on the cell line or sample type) would be helpful. What do the histograms for 'ideal' experimental parameters look like? What quantitative metrics were used to optimize experimental conditions and the protocol for this technique? This kind of description could be helpful to researchers looking to adopt the assay for different types of samples.

d. What were the minimum and maximum times for suspension in DNA intercalator solution after antibody labelling?

e. No culture conditions for the HEK 293T cells used in Fig. S2 are described.

f. The authors should state how gates and thresholds were chosen in their analysis to improve understanding by broad audiences (e.g. "the frequency of cells with greater than 10 counts of IdU was used to quantify DNA synthesis activity for each immune-cell phenotype cluster" vs. the >20 counts used in Fig. S2 to identify "frequency of cells positive for IdU").

g. The authors state that 5-10x10⁶ cells from each condition were stained for analysis in the two types of primary sample studies, but that each dataset contained ~1-3x10⁵ cells. The in silico isolation gating for peripheral immune mononuclear cells appears to pass ~16% (Fig. S3) of the initial population (leading to 8x10⁵ cells/condition if my calculations are correct) while that for the bone marrow studies appears to pass ~76% (Fig. S4) and ~73% (Fig. S5) of the initial population. Are there additional cell loss or sample splitting steps not clear from the described Methods?

h. The caption for the gating scheme of Fig. S3 would benefit from additional information on the methodology used, to aid in understanding by broad audiences.

6. What is the puromycin signal in the CD4+ T cell population in the unlabeled "No SOM3B" group of Fig. 3c?

Minor notes:

1. The resolution of Fig. 5 in the review manuscript is very poor, hindering readability of the text.

2. Arrows/arrowheads depicting points of interest in figures should be described in the figure caption if possible, as well as in the main text.

3. Ref. 10 in the supplementary information does not contain the full author list. References should be double-checked for accuracy.

4. Several typographical errors remain in the manuscript after the suggested proofreading by prior Reviewer 3. A more thorough proofreading should be performed. Examples:

a. "labeling SOM3B labeling" p.24

b. "staining is performed in the absence inhibitors" p.5

c. "events from two population under comparison" supplementary p.8

d. "arsinh" supplementary p.6

e. "with by SPADE" Fig. S5 caption

Reviewers' comments

Author responses

Changes from previous version highlighted in manuscript text

Reviewer #3 (Remarks to the Author):

The authors have presented the first demonstration of multiplexed measurement of DNA replication, RNA transcription, and protein synthesis alongside tens of additional biomolecule targets via pulsed labelling and antibody probing, all measured in parallel from single cells using mass cytometry. The impressive multiplexing enabled by mass cytometry and the authors' labelling strategy creates a useful tool to understand cellular dynamics at the single-cell level, and the authors have demonstrated its utility by showing its application to understanding cell cycle dynamics of human cell lines as well as stimulated whole blood mononuclear cells and the hematopoietic continuum from clinical samples. While this kind of labelling technique has the potential to open up new studies understanding regulation processes in biological systems, the paper would benefit from additional characterization/validation of the technique in order to improve its adoption and reproducibility by other researchers.

We thank the reviewer for this characterization of our tri-molecule labeling methodology and its combination with mass-cytometry reagents. We agree that the combination of labeling molecules used in our method with multiplexing of mass-cytometry will uncover new facets of dynamic biomolecule synthesis at the single-cell level when applied to other cell- or organ-systems outside of those presented in our manuscript. Further, we thank the reviewer for a close inspection of the manuscript and for providing thorough suggestions to materially improve the manuscript and data presentation for readability and reproducibility by other researchers. Along with providing responses to Reviewer #3's suggestions, we improved the readability and references of the manuscript to promote clarity, reproducibility, and adoption by other researchers.

The major claim of the paper is the first demonstration of simultaneous single-cell measurement of de novo DNA, RNA, and protein molecules, alongside multiplexed detection of a suite of other biomolecules, allowing insight into biomolecule synthesis and dynamics. Claims are novel and paper would be of interest. This kind of multiplexed measurement will aid in the design of studies to better understand how de novo DNA, RNA, and protein synthesis is related to phenotype and cell state across biological systems, as well as the dynamics of cell response to stimulus. Several of the points raised by the prior reviewers in the previous version of the manuscript (specifically related to quantitative analysis of the data: prior Reviewer 3's question regarding whether a bootstrapping approach like that used in Fig. 3 to estimate confidence intervals could also be used across the remaining datasets of the work). The paper would benefit from additional quantitative analysis to support some of the claims (please see suggested revisions below). This quantitative analysis would strengthen the key claims of the paper and improve adoption and reproducibility by other researchers. A clearer description of what has and has not been previously characterized and validated with respect to the labelling molecules used in this work would, along with the authors' own

characterization, demonstrate solid validation of the technique.

While the manuscript is clearly written, some sections could benefit from additional explanation to aid in understanding by and impact on the broad audience of this journal (e.g.: previous work on and validation of these labeling molecules; capabilities and limitations on quantification and comparisons between different sample and cell types using this labelling technique). Additional statistics and quantification to back up key claims would be beneficial (see suggestions below). A brief overview of the specific labelling molecules used in the work (or very similar molecules) and their previous application would improve both the completeness of previous literature citation as well as background basis for the work (e.g. what validation of these techniques and molecules has already been performed by others vs. what needed to be performed in this contribution). This could be included either in the background section or at the beginning of the results section, where the molecules are introduced.

Again, we wish to thank the reviewer for their provided comments and suggestions, specifically to expand upon previous applications of metabolic pre-cursor molecules utilized in our study. Accordingly, we have expanded the results section where we introduce the three molecules that are used to monitor DNA, RNA, and protein synthesis, respectively. Additionally, we expanded the discussion section to add commentary of the importance of including control samples when performing SOM₃B labeling, both to confirm the validity of custom conjugated antibody reagents and to observe and optimize conditions to minimize background signal of incorporated molecules when applied to new cell types. Importantly, we believe that with the additional information provided, specifically regarding historical use of the metabolic precursor molecules which comprise the SOM₃B labeling reagents, and commentary on the widespread use of these molecules in many areas of bioscience research, will provide adequate information for the adoption of this technique by researchers previously unfamiliar with their application. Finally, we performed additional statistical testing to compare incorporation of single, double, or triple-labeling with SOM₃B reagents for Jurkat cell experiments and primary whole blood experiments, as outlined below, in order to quantitatively assess the incorporation of these molecules when combined and their effect on cellular fitness.

Suggested revisions:

1. As the primary contribution of this work is the tri-molecular labelling and quantification strategy for single cells, the reproducibility and quantitative detection thresholds for this strategy need to be assessed. I could not see any sample-to-sample measurement variability metrics outlined in the manuscript, and an understanding of this variability is crucial to drawing conclusions from the data presented. Detection thresholds for significant changes in measured metrics need to be defined in order to conclude that “simultaneous labeling and detection of cells with IdU, BRU, and puromycin did not interfere with quantification or cell signaling activity when compared with single- or double-molecule labeling controls”, that “Bone marrow cell fitness and surface protein expression are unperturbed by labeling with SOM₃B reagents”, or that “SOM₃B does not interfere with primary immune sample cell fitness”. Indeed, the plots and presented data suggest these conclusions, but the authors’ claims would be strengthened by a quantitative analysis showing that any changes are within the measurement variability of the system. In several figures (e.g. Figs. 1, 2, 3, 4, S1, S2) the authors state that experiments were

performed multiple times, but statistical analysis of this replicate data would support the authors' conclusions and strengthen their validation of this technique.

We thank the reviewer for their close inspection of the presented plots and for the recommendation to provide more quantitative analysis on data presented in the manuscript. It was our goal to present raw data when possible in order for the audience to appreciate the single-cell nature of the biosynthesis activities SOM₃B is capable of producing, leading us to present single-cell data in histogram and biaxial plot formats where appropriate. Importantly, experiments were repeated to ensure reproducibility and robustness of SOM₃B method across several cell types, including the cell lines and primary samples presented in this manuscript, as well as many other cell systems yet to be published. However, these experiments were not conducted in a way (or with the goal) to statistically/quantitatively compare biosynthesis activity across individual repeats. This is because small variations in cell number, SOM₃B labeling time, or independent mass-cytometry acquisition conditions can induce variation in the data independent of the methods. For this reason, a number of our descriptors have been purposefully qualitative to general observations where quantitative/statistical comparisons are limited to cells within a given experiment.

Thus, for all experimental data for which we provide comparison across experimental conditions, (Fig 1b, c, d, e; Fig S1d; Fig S2 a, b; Fig 3 & Fig 4) or within subpopulations (Fig 5, Fig S6), we use a barcoding approach (i.e. palladium-isotope¹) which allowed us to pool experimental samples after they have been fixed with paraformaldehyde (as single-cell suspensions), into a single vessel, followed by staining with isotope conjugated antibodies used to quantify biosynthesis activity and other measures of cell fitness, phenotype, signaling, etc. This step is critical to eliminate sample-to-sample staining variation otherwise introduced when preparing samples in separate staining tubes (i.e. incubating with isotope-labeled antibodies for puromycin, BRU, etc.), and enables the simultaneous acquisition of all experimental conditions on the mass-cytometer in a single acquisition run. Accordingly, we performed additional quantitative analysis for this current revision in order to more statistically/quantitatively compare experimental samples collected in this way, and expand upon these analyses below.

a. The data of Fig. S1 (crucial validation of the ability to perform the simultaneous measurements key to the claims of this work) would benefit from a quantitative statistical analysis to show that each measurement does not interfere with the others, or with cell signaling measurements (to support the statement “simultaneous labeling and detection of cells with IdU, BRU, and puromycin did not interfere with quantification or cell signaling activity when compared with single- or double-molecule labeling controls”. The inhibitor validation data of Fig. S2 would also benefit from a similar quantitative analysis of dose-dependent effects (while the authors describe changes in the median BRU/puromycin incorporation I did not see any quantification of these medians or their uncertainty). These additional quantitative assessments would provide key validation for the primary novel contribution of this work.

Using the methods established in the previous revision (i.e. comparisons in Figure 5), we performed testing with the Freidman-Rafsky test statistic to demonstrate that combining the three labeling molecules does not interfere with other synthesis processes, nor does it interfere with cell signaling. We also added raw median values to provide quantification of the dose-dependent response in biosynthesis activities as a result of pre-exposure to small molecule inhibitors. This

is presented in revised Fig. S1 in this re-submission and has not changed the conclusions presented in the original manuscript.

Further, we added a comment to the discussion section on the importance of including adequate controls and to validate incorporation with small molecules inhibitors, a common practice for researchers in the cell-biology field, but which may not be evident to researchers in other fields.

b. The data of Fig. 3 would also benefit from quantitative statistical analysis to support the claim of “control samples from the same PBMC donor left unlabeled, or pre-treated with specific

inhibitors, indicated minimal effect on cell fitness as there were little change in the proportion of cells with activated pro-apoptotic caspase molecules or in cell viability, examined by cisplatin exclusion” (p.9). A similar comment for p.10 (Fig. S4): “we confirmed labeling primary human bone marrow with SOM3B reagents did not interfere with cellular fitness (Fig. S4)...”.

We performed quantitative statistical comparison of the viability and apoptotic status between un-labeled and SOM₃B labeled whole blood presented in **Figure 3**, and bone marrow presented in **Figure S4**, with results above. Importantly, we do not observe any significance between these cellular fitness characteristics when whole blood is exposed to SOM3B labeling reagents or left unlabeled and this analysis has not changed the conclusions presented in the original manuscript.

2. In Fig. 2 the IdU colour ranges for the two cell lines differ dramatically (~5x greater max for the HeLa than the Jurkat). Does this allow us to draw any information about the cell types in question, or is higher labelling with the three molecules expected for some cells compared to others for reasons unrelated to cellular synthesis rates? Does this present any implications for

quantification of diverse clinical sample populations or assay optimization?

We thank the reviewer for pointing out this subtle difference between the two presented cancer cell lines. In fact, HeLa cells demonstrate a shortened S-phase length than Jurkat cells (8.8 hours for HeLa², 10-16 hours for Jurkat³). Thus, HeLa cells would thus need to operate DNA synthesis machinery at a higher flux of nucleosides into the synthesized DNA, resulting in a higher median of incorporated IdU for S-phase cells when exposed to IdU at the same concentration and duration relative to Jurkat cells. While this type of comparison across cell lines is an interesting future application of SOM₃B, the intention of data presented in Figure 2 was to illustrate the capability to recover dynamic biosynthesis states across a single sample (i.e. DNA synthesis in S-phase cells and suppression of RNA and protein synthesis in mitosis). We edited the results section of cell cycle data to reflect this.

3. P.6: The authors state that the data of Fig. S2, showing an effect of cycloheximide treatment on transcription and DNA synthesis, suggest “a dependence of DNA and RNA activity on continued protein synthesis for this cell line”. Additional support (by referencing of past studies investigating effects of cycloheximide on transcription and DNA synthesis) would help support this conclusion on the mechanism driving the observation.

Relevant references supporting our conclusion of cycloheximide effect on transcription⁴ and DNA synthesis⁵ were added to the main text.

4. For the stimulation experiment, should there be a control for the time between blood draw/brefeldin A treatment (permitting intracellular detection of cytokines) and initiation of stimulation? If not, please provide a rationale and reference(s).

*We thank the reviewer for inquiring on inclusion of control conditions for the stimulation experiment. Brefeldin A is a widely used reagent to measure intracellular production of cytokines which would otherwise be released⁶. Further, it has been widely utilized with in vitro, in vivo and ex vivo studies. We would like to point the reviewer back to the detailed methods section, as it appears there is some confusion on the timing of addition of brefeldin A with respect to the stimulatory agents (PMA/Ionomycin). Importantly, all six samples prepared for the time-course **received brefeldin A at the same time** (i.e. t=0h), with each time course sample subsequently receiving stimulatory agent according to the time-course series, and the final “No stimulation” sample not receiving any stimulation agent. The experiment ends with labeling with SOM₃B reagents during the final 30 minutes, and all time course samples are then simultaneously lysed of red blood cells, and fixed with formaldehyde using the commercially available reagent cited in the detailed methods. Thus, for all time course samples, they were exposed to brefeldin A for the same duration, with the only differing variable being the time stimulatory agent was added before labeling and fixation. Such an experiment will thus reflect dynamic changes of de novo biosynthesis, phosphorylation, and biomolecule abundance as a result of exposure to stimulatory agent.*

5. Additional experimental details would be beneficial to ensure reproducibility and adoption of this technique:

a. On page 5, the authors state: “After as little as 30 minutes, the labeling mixture is removed and

cells can be fixed or cryopreserved for later analysis”, but cryopreservation prior to fixation is not described in the methods (methods state “After fixation at room temperature for 15 minutes, cells were washed with DPBS twice, re-suspended in CSM and stored at -80C until antibody staining.”). Were some samples cryopreserved prior to fixation? If so, in what types of experiments is this the preferred strategy vs. immediate fixation?

We thank the reviewer for pointing out ambiguity with sample storage prior to antibody labeling. The first sentence was edited to reflect that samples can be fixed and prepared for analysis immediately after, or cryopreserved (stored at -80C) after fixation for later antibody staining and data acquisition. No data presented is derived from samples that were cryopreserved before fixation, as this is not a storage method we explicitly tested.

b. The methods should include product numbers, particularly for the antibodies in Tables S1 and S2. Although manufacturer, target, and clone of each antibody are all presented in these tables, specific product numbers and potentially lot numbers would be beneficial for reproducibility of this antibody-reliant study.

We thank the reviewer for their raised concern regarding inclusion of product and lot numbers. In our revised manuscript, we included product numbers for previously unreported metal-labeled monoclonal antibody reagents (e.g. anti-Puromycin and anti-BRDU). While providing additional information on all other antibodies used in this study, such as the product and lot numbers, may appear to improve reproducibility to researchers unfamiliar with immunohistochemistry, we believe that the level of antibody reporting provided in our manuscript is sufficient for reproduction of the experiments performed in our study. Importantly, all other antibodies against phenotype and functional markers utilized in our study have previously been described in literature for mass-cytometry applications⁷⁻¹¹. Finally, manufacturer information such as product numbers and lot numbers that are subject to change, and thus will subsequently render this information irrelevant. However, target antigen and clone number are likely to remain constant with monoclonal antibody producing cell lines.

c. The authors should describe how the optimal antibody concentrations were chosen from the data in Fig. S1a-b. Similarly, a brief summary of how the optimal BRU/puromycin concentrations were chosen based on the data of Fig. 1 b-c (and potentially whether there is any variation in optimal concentration based on the cell line or sample type) would be helpful. What do the histograms for ‘ideal’ experimental parameters look like? What quantitative metrics were used to optimize experimental conditions and the protocol for this technique? This kind of description could be helpful to researchers looking to adopt the assay for different types of samples.

We thank the reviewer for addressing how concentrations of reagents were selected for labeling cells with metabolic precursor molecules and metal-labeled antibodies. Accordingly, we provided additional detail regarding these decisions in the results section where the use of the three precursor molecules is first introduced, and to the antibody labeling section of the detailed methods.

In brief, values of precursor molecule concentration were selected by performing titrations of the precursor molecule itself, and all stained with the same antibody concentration

for all pre-cursor molecule concentrations tested, including a sample of unlabeled cells. This titration allows the observation of any background binding of metal labeled antibodies with inspection of the single-cell data belonging to the unlabeled condition. As there is little detectable background signal of either the metal labeled BRU or puromycin antibody, we conclude the antibody concentration used is sufficient to only stain incorporated molecules. Next, we selected a concentration of pre-cursor molecule which would enable a dynamic range of activity, and thus allow the assessment of variable activity of these processes within a single sample. This procedure is also followed for determination of antibody concentration, comparing concentrations on the same sample of unlabeled or labeled cells.

d. What were the minimum and maximum times for suspension in DNA intercalator solution after antibody labelling?

We thank the reviewer for addressing the duration of DNA intercalator solution cells are kept in before acquisition on the mass-cytometer. Detailed protocols regarding metal-labeled antibody staining could previously be found in references 1, 2, 3, and 10 in the originally submitted manuscript in the detailed methods section (and references 7-11 in this response). We have added to the detailed methods section that samples were stained with intercalator for 20 minutes at room temperature before acquisition on the same day, or overnight at 4C for acquisition on the day following staining. For mass cytometry it is widely accepted that the intercalator is used at sub-saturating concentration – thus the staining volume / cell number / concentration are more important than staining time as peak staining with the high affinity reagent is reached quite quickly.

e. No culture conditions for the HEK 293T cells used in Fig. S2 are described.

We thank the reviewer for addressing the culture conditions used for HEK 293T cells that were used in figure S2. In the original manuscript, and the second revision, a sentence can be found at the end of the “Cell Culture” section stating that Human Embryonal Kidney (HEK) cells were cultured under the same conditions used to culture HeLa cells. In the revised manuscript, we moved this sentence to the top of the “Cell Culture” section and included ‘293T’ incase this sentence was missed due to its placement at the end of this section of the detailed methods.

f. The authors should state how gates and thresholds were chosen in their analysis to improve understanding by broad audiences (e.g. “the frequency of cells with greater than 10 counts of IdU was used to quantify DNA synthesis activity for each immune-cell phenotype cluster” vs. the >20 counts used in Fig. S2 to identify “frequency of cells positive for IdU”).

As DNA pre-cursor molecules are used routinely by cell biologist to quantify DNA synthesis and/or proliferation, it is common practice to perform a control experiment on unlabeled cells to determine the extent of background signal before selecting thresholds/gates for cell populations positive for incorporation. Accordingly, we added this information to the “Data Analysis” section of the detailed methods stating such.

g. The authors state that 5-10x10⁶ cells from each condition were stained for analysis in the two

types of primary sample studies, but that each dataset contained $\sim 1-3 \times 10^5$ cells. The in silico isolation gating for peripheral immune mononuclear cells appears to pass $\sim 16\%$ (Fig. S3) of the initial population (leading to 8×10^5 cells/condition if my calculations are correct) while that for the bone marrow studies appears to pass $\sim 76\%$ (Fig. S4) and $\sim 73\%$ (Fig. S5) of the initial population. Are there additional cell loss or sample splitting steps not clear from the described Methods?

Previously submitted versions of our manuscript did indeed include description of cell processing of primary samples. Importantly, primary whole blood and bone marrow contain a large relative proportion granulocyte cells (<https://www.biolegend.com/newsdetail/frequencies-blog/5396/>) which were not are not the focus of this study. Thus, we employed reagents enabling selective removal of this cell type for bone marrow data in order to allow isolation of rare, mononucleated progenitor populations. We revised the methods of the manuscript surrounding the use of simultaneous lyses and fixation buffer in the main text, with a pointer for readers to the detailed methods section where these reagents and procedure are explained.

h. The caption for the gating scheme of Fig. S3 would benefit from additional information on the methodology used, to aid in understanding by broad audiences.

In the revised manuscript, we included a detailed description of features for mononuclear cell isolation in the caption of Fig S3, including removal of debris, non-viable cells, and phenotype descriptions of removed cell types. Additionally, we point readers to Table S3 which provides further detail on cell phenotype and relevant references.

6. What is the puromycin signal in the CD4+ T cell population in the unlabeled “No SOM3B” group of Fig. 3c?

The signal observed for puromycin in the CD4+ population is a result isotopic contamination in the CD4-isotope labeled antibody from the adjacent isotopic mass, which in our study is used to label the puromycin antibody. CD4 is labeled with Gadolinium-157, which contains about 5% Gadolinium-158. Observation of isotopic contamination/bleed is not uncommon and 5% of 157 in 158 is the worst case for all mass cytometry reagents used here. The plot below highlights the source of the signal in untreated (no puromycin) cells. Positive signal is only observed in cells with the highest expression of CD4 (red arrow). While this can be confusing, it is unrelated to the SOM₃B method itself and instead related to the mass cytometry readout.

For clarification, we added a statement to the Fig 3c caption to explain this signal, and further highlight the importance of including adequate controls in experiments using SOM₃B to optimize reagents and assist in panel construction.

Minor notes:

1. The resolution of Fig. 5 in the review manuscript is very poor, hindering readability of the text.
2. Arrows/arrowheads depicting points of interest in figures should be described in the figure caption if possible, as well as in the main text.
3. Ref. 10 in the supplementary information does not contain the full author list. References should be double-checked for accuracy.
4. Several typographical errors remain in the manuscript after the suggested proofreading by prior Reviewer 3. A more thorough proofreading should be performed. Examples:
 - a. “labeling SOM3B labeling” p.24
 - b. “staining is performed in the absence inhibitors” p.5
 - c. “events from two population under comparison” supplementary p.8
 - d. “arsinh” supplementary p.6
 - e. “with by SPADE” Fig. S5 caption

All minor corrections as per the reviewer recommendations, have been made in the revised manuscript where indicated

Reference:

1. Zunder, E. R. *et al.* Palladium-based mass tag cell barcoding with a doublet-filtering scheme and single-cell deconvolution algorithm. *Nat. Protoc.* **10**, 316–33 (2015).
2. Hahn, A. T., Jones, J. T. & Meyer, T. Quantitative analysis of cell cycle phase durations and PC12 differentiation using fluorescent biosensors. *Cell Cycle* **8**, 1044–1052 (2009).
3. Eidukevicius, R. *et al.* A method to estimate cell cycle time and growth fraction using bromodeoxyuridine-flow cytometry data from a single sample. *BMC Cancer* **5**, 1–11 (2005).
4. Willems, M., Penman, M. & Penman, S. The regulation of RNA synthesis and processing in the nucleolus during inhibition of protein synthesis. *J. Cell Biol.* **41**, 177–187 (1969).
5. Noy, G. P. & Weissbach, A. HeLa cell DNA polymerases: The effect of cycloheximide in vivo and detection of a new form of DNA polymerase α . *Biochim. Biophys. Acta - Nucleic Acids Protein Synth.* **477**, 70–83 (1977).

6. Yin, Y., Mitson-Salazar, A. & Prussin, C. Detection of intracellular cytokines by flow cytometry. *Curr. Protoc. Immunol.* **2015**, 6.24.1-6.24.18 (2015).
7. Behbehani, G. K., Bendall, S. C., Clutter, M. R., Fantl, W. J. & Nolan, G. P. Single-cell mass cytometry adapted to measurements of the cell cycle. *Cytom. Part A* **81A**, 552–566 (2012).
8. Bendall, S. C. *et al.* Single-cell mass cytometry of differential immune and drug responses across a human hematopoietic continuum. *Science* **332**, 687–696 (2011).
9. Bodenmiller, B. *et al.* Multiplexed mass cytometry profiling of cellular states perturbed by small-molecule regulators. *Nat. Biotechnol.* **30**, 858–67 (2012).
10. Bendall, S. C. *et al.* Single-Cell Trajectory Detection Uncovers Progression and Regulatory Coordination in Human B Cell Development. *Cell* **157**, 714–725 (2014).
11. Good, Z. *et al.* Single-cell developmental classification of B cell precursor acute lymphoblastic leukemia at diagnosis reveals predictors of relapse. *Nat. Med.* **24**, 474–483 (2018).